# Polybasic Speculative Decoding Through a Theoretical Perspective

**Ruilin Wang** [1]   **Huixia Li** [2]   **Yuexiao Ma** [1 2]   **Xiawu Zheng** [1 3 4]   **Fei Chao** [1]   **Xuefeng Xiao** [2]   **Rongrong Ji** [1 3]

## Abstract

Inference latency stands as a critical bottleneck in the large-scale deployment of Large Language Models (LLMs). Speculative decoding methods have recently shown promise in accelerating inference without compromising the output distribution. However, existing work typically relies on a dualistic draft-verify framework and lacks rigorous theoretical grounding. In this paper, we introduce a novel *polybasic* speculative decoding framework, underpinned by a comprehensive theoretical analysis. Specifically, we prove a fundamental theorem that characterizes the optimal inference time for multi-model speculative decoding systems, shedding light on how to extend beyond the dualistic approach to a more general polybasic paradigm. Through our theoretical investigation of multi-model token generation, we expose and optimize the interplay between model capabilities, acceptance lengths, and overall computational cost. Our framework supports both standalone implementation and integration with existing speculative techniques, leading to accelerated performance in practice. Experimental results across multiple model families demonstrate that our approach yields speedup ratios ranging from $3.31\times$ to $4.01\times$ for LLaMA2-Chat 7B, up to $3.87\times$ for LLaMA3-8B, up to $4.43\times$ for Vicuna-7B and up to $3.85\times$ for Qwen2-7B—all while preserving the original output distribution. We release our theoretical proofs and implementation code to facilitate further investigation into polybasic speculative decoding.

---

[1]Key Laboratory of Multimedia Trusted Perception and Efficient Computing, Ministry of Education of China, Xiamen University, 361005, P.R. China [2]ByteDance [3]Institute of Artificial Intelligence, Xiamen University [4]Peng Cheng Laboratory, Shenzhen, China. Correspondence to: Rongrong Ji <rrji@xmu.edu.cn>.

*Proceedings of the $42^{nd}$ International Conference on Machine Learning*, Vancouver, Canada. PMLR 267, 2025. Copyright 2025 by the author(s).

## 1. Introduction

Large Language Models (LLMs) have substantially advanced Natural Language Processing (NLP), achieving leading performance in a wide range of tasks. Yet their exceptional capabilities are tempered by significant computational demands, particularly in low-latency scenarios. Among multiple acceleration techniques, *speculative decoding* (Stern et al., 2018; Leviathan et al., 2023; Xia et al., 2023; Chen et al., 2023a; Anonymous, 2025b;a; Teng et al., 2024; Zhang et al., 2024b) has emerged as a strategy to speed up inference while preserving output fidelity.

The current speculative decoding ecosystem largely hinges on *draft-then-verify* paradigms, which spawn various sub-directions such as the design of lightweight draft models (Leviathan et al., 2023; Xia et al., 2023; Chen et al., 2023a; Kim et al., 2024a; Svirschevski et al., 2024; Yin et al., 2024; Sadhukhan et al., 2024), hierarchical token structures (Stern et al., 2018; Miao et al., 2024; Du et al., 2024), and unified architectures (Yi et al., 2024; Cai et al., 2024). Verification strategies typically follow three approaches: greedy sampling, speculative sampling (Leviathan et al., 2023), and typical acceptance (Cai et al., 2024). Despite these efforts, current strategies remain limited by a **dualistic** relationship between draft and target models (Qin et al., 2024; Liu et al., 2024; Gui et al., 2024; Khisti et al., 2024), affecting key parameters such as acceptance length due to inherent capacity disparities between the two models. While some recent works (Chen et al., 2023b; Kim et al.; Spector & Ré, 2023) investigate multi-level drafts, they still employ a singular top-level target model. Moreover, the field has hitherto lacked an overarching theoretical framework to guide system design and provide robust performance guarantees.

In this paper, we introduce a principled **polybasic** speculative decoding framework that uses multiple interconnected models, grounded in a thorough theoretical analysis. Our investigation yields two central insights. First, we derive a fundamental relationship between the number of forward passes and average acceptance lengths that dictates optimal system-level inference speed. This relationship allows us to precisely quantify potential inference speedups when adding additional models. Second, we establish the capacity of *speculative sampling* to enhance stability in acceptance lengths. By optimizing sampling parameters, we can reduce

variance in token acceptance to achieve more predictable performance.

Building on these insights, we propose a unified architecture for polybasic speculative decoding wherein multiple draft models coordinate with each other and with a single target model. Our implementation guidelines detail how to select models, set speculation lengths, and implement multi-stage verification procedures that maximize throughput. Experimental evaluations demonstrate that this approach outperforms typical dualistic methods on diverse tasks including MT-bench (Zheng et al., 2023), translation, summarization, QA, mathematical reasoning, and Retrieval-Augmented Generation (RAG). Our empirical results indicate that the proposed polybasic framework maintains output fidelity while delivering speedup ratios from $3\times$ to over $4\times$ across a range of widely used LLMs (e.g., Vicuna-7B, LLaMA2-Chat 7B, LLaMA3-8B).

The main contributions of this work are summarized as follows:

- We develop a formal theoretical framework for polybasic speculative decoding, identifying the system-level dependencies between model forward-pass cost, acceptance lengths, and stable acceleration performance.

- We prove a fundamental theorem that provides a rigorous expression for optimal inference time in multi-model speculative decoding, highlighting conditions under which additional models improve speedups.

- We show that speculative sampling significantly reduces variance in token acceptance lengths for multi-model settings, increasing stability and improving inference throughput.

- Our empirical investigation demonstrates the effectiveness of the proposed polybasic approach, achieving notable speedups (up to $4.43\times$) on widely used LLMs across various tasks, while preserving the target model's output distribution.

## 2. Related Work

The concept of speculative decoding originated from blockwise parallel decoding (Stern et al., 2018), showcasing the viability of partially parallel language generation. More recently, research on speculative decoding (Stewart et al., 2024; Zafrir et al., 2024) has coalesced around two dimensions: *drafting* strategies for token prediction and *verification* mechanisms for ensuring correctness.

**Drafting Strategies.** Drafting approaches typically follow either *independent* or *self-drafting* protocols. *Independent drafting* involves utilizing smaller or more efficient models

to propose candidate tokens, later verified by a larger model. Methods range from training specialized drafters (Leviathan et al., 2023; Xia et al., 2023; Chen et al., 2023a; Kim et al., 2024a; Metel et al., 2024) to zero-shot usage of pre-existing models (Spector & Ré, 2023). *Self-drafting* employs the same model at intermediate stages, as in blockwise decoding (Stern et al., 2018; Xiao et al., 2024), early exiting (Yang et al., 2023), or mask-predict (Zhao et al., 2024), aiming to amortize computation within the same architecture.

**Verification Mechanisms.** Verification primarily ensures that proposed tokens maintain consistency with the target distribution. Greedy verification (Kim et al., 2024a; Xia et al., 2023; Agrawal et al., 2024) is conceptually straightforward but may hinder speedups for certain tasks. Speculative sampling (Leviathan et al., 2023) introduces a probabilistic acceptance rule that adaptively filters tokens while retaining a high acceptance length. Token-tree-based verification (Miao et al., 2024; Spector & Ré, 2023; Lu et al., 2024; Gao et al., 2024) provides hierarchical checks, which can be beneficial for highly parallel architectures.

**Recent Advances and Limitations.** Recent work on cascade or multi-level drafting (Chen et al., 2023b; Sun et al., 2024) has partially moved beyond the dualistic draft-target scheme. TRIFORCE (Sun et al., 2024) tackles long sequence generation by introducing a two-level hierarchy with retrieval-based drafting and partial KV cache as an intermediate layer, achieving up to $2.31\times$ speedup for Llama2-7B-128K. CS Drafting (Chen et al., 2023b), on the other hand, employs vertical and horizontal cascades to eliminate neural autoregressive generation and optimize time allocation in drafting, resulting in up to $81\%$ additional speedup over standard speculative decoding. While TRIFORCE focuses on memory efficiency in long-context scenarios through KV cache optimization, CS Drafting targets general inference optimization through cascade structures and statistical drafting.

However, these approaches usually rely on empirical heuristics without a unified theoretical framework to guide model selection, acceptance-length control, and stability analysis. Our work explicitly addresses these gaps by introducing a comprehensive theoretical treatment of *polybasic* speculative decoding and validating the resulting system design empirically.

## 3. Polybasic Speculative Decoding Framework

Although speculative decoding has been demonstrated as an effective technique for single-model verification, its acceleration potential remains capped by the inherent draft-target capacity gap in dualistic paradigms. We propose a *polybasic* speculative decoding framework, which systematically

**(a) Dualistic speculative decoding**     **(b) Polybasic speculative decoding**

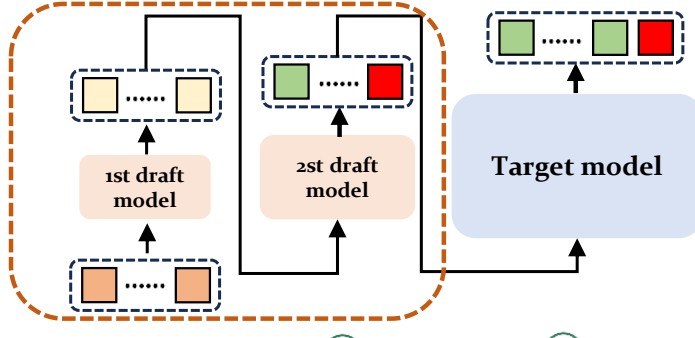

*Figure 1.* Comparison of speculative decoding frameworks. (a) Traditional dualistic approach with a single draft model. (b) Our polybasic framework with multiple draft models achieves superior performance (4× speedup and 8-10 tokens acceptance length) while maintaining good generalization ability. The framework demonstrates significant improvements over the dualistic baseline.

employs multiple models to increase parallelism and acceptance length while preserving fidelity to the final target distribution. Below, we detail the core problem setting, theoretical foundations, and practical instantiations.

### 3.1. Problem Formulation

Let us consider a chain of models $\mathcal{M} = \{M_1, \ldots, M_n\}$, where $M_1$ is the final target model we wish to replicate in distribution, and $M_2, \ldots, M_n$ act as drafters at progressively lower capacity (higher index indicates a smaller or faster model). Let $\mathcal{V}$ be the vocabulary, and $p_i(x \mid x_{\leq t}) = M_i(x_{\leq t})$ the distribution over $\mathcal{V}$ given context $x_{\leq t}$.

At each decoding step $t$, $M_n$ drafts a block of $K$ tokens, verified in ascending order by $M_{n-1}, M_{n-2}, \ldots, M_1$. Tokens are accepted if they do not exceed a certain mismatch criterion, reflecting *speculative sampling*, greedy matching, or another verification rule. Define:

$$L_i = \mathbb{E}\big[(\text{\# of consecutive tokens accepted by } M_i)\big], \quad (1)$$

i.e., the expected block length accepted when verifying with model $M_i$. Denoting by $F_i$ the number of forward passes that $M_i$ must perform, our goal is to minimize the total inference time

$$T = \sum_{i=1}^{n} F_i \cdot T_i, \quad (2)$$

where $T_i$ is the cost of a single forward pass for model $M_i$.

### 3.2. Theoretical Foundations

We establish fundamental properties of multi-model (polybasic) speculative decoding that govern how additional models

impact computational cost and acceptance lengths. Our analysis focuses on two main aspects: *(i) optimal inference time* and *(ii) stability of acceptance lengths*.

**Optimal Inference Time.** In a conventional dualistic system with one draft model ($M_2$) and a single target ($M_1$), the total inference time is approximately equal to $\frac{N}{L_1} T_1 + \beta \frac{N}{L_1} T_2$, where $N$ is the sequence length and $\beta$ is a system-dependent scaling factor reflecting the final draft model's capability. In a polybasic setting with $n > 2$ models, additional drafting layers can bring more tokens per verification cycle if acceptance lengths between intermediate pairs are high. However, each additional model also introduces its own forward-pass costs. Formally, we have:

**Lemma 3.1** (Optimal Inference Time). *For an $n$-model polybasic system generating $N$ tokens, the total inference time $T$ can be expressed as:*

$$T = \sum_{i=1}^{n-1} \frac{N}{L_i} \cdot T_i + \beta \cdot \frac{N}{L_{n-1}} T_n, \quad (3)$$

*where $L_i$ is the expected acceptance length for verification by $M_i$, and $\beta$ is a system-dependent scaling factor reflecting the final draft model's capability.*

*Sketch of Proof.* We segment the total generation length $N$ into accepted blocks validated by pairs $(M_i, M_{i+1})$. Each model $M_i$ must run as many forward passes as needed to accept $N$ tokens in total. A more detailed version of the proof incorporates the block acceptance process for each adjacency $(M_i, M_{i+1})$, culminating in the time decomposition of Equation (3). $\qquad\square$

**Model Selection Criterion.** The next question is whether introducing a new model $M_{\text{new}}$ between $M_i$ and $M_{i+1}$ improves $T$. The improvement depends on whether the reduced cost from higher acceptance length outweighs the additional forward-pass overhead. Formally:

**Theorem 3.2** (Model Insertion Efficiency). *Adding $M_{\text{new}}$ between $M_i$ and $M_{i+1}$ decreases total inference time if and only if it achieves a sufficiently large increase in acceptance lengths, balanced against its forward-pass cost $T_{\text{new}}$. Concretely, if $L_{\text{new}}$ is the acceptance length when verifying tokens from $M_{\text{new}}$ against $M_i$, and $L'_{i+1}$ is the acceptance length from $M_{i+1}$'s perspective, then improvement occurs if:*

$$\frac{T_{\text{new}}}{T_i} < L_{\text{new}}\left(\frac{1}{L_i} - \frac{1}{L_{i-new}}\right) \quad or$$
$$\frac{T_{\text{new}}}{T_{i+1}} < \beta\left(\frac{L_{\text{new}-(i+1)}}{L_i} - 1\right)$$

*Proof.* First, we prove the case for three models:

For $i = 2$:
$$T = \frac{N}{L_1} \cdot T_1 + \beta \cdot \frac{N}{L_1} \cdot T_2 \tag{4}$$

For $i = 3$:
$$T = \frac{N}{L'_1} \cdot T_1 + \frac{N}{L'_2} \cdot T'_2 + \beta \cdot \frac{N}{L'_2} \cdot T'_3 \tag{5}$$

where $T_i$ is the inference time of the $i$-th model, $\alpha$ is considered to be equal in both equations, and $T_2 = T'_3$, $L'_1 > L'_2 > L_1$.

We can calculate the difference between Equation 4 and Equation 5:

$$N \cdot \left(\frac{1}{L'_1} - \frac{1}{L_1}\right) \cdot T_1 + \frac{N}{L'_2} \cdot T'_2$$
$$+ \beta \cdot N \cdot \left(\frac{1}{L'_2} - \frac{1}{L_1}\right) \cdot T_2 < 0$$

The expression is less than 0 if either of the following conditions is met:

Condition 1: Sum of the first two terms is less than 0

$$N \cdot \left(\frac{1}{L'_1} - \frac{1}{L_1}\right) \cdot T_1 + \frac{N}{L'_2} \cdot T'_2 < 0$$
$$\Leftrightarrow \frac{T'_2}{T_1} < L'_2 \cdot \left(\frac{1}{L_1} - \frac{1}{L'_1}\right)$$

OR

Condition 2: Sum of the last two terms is less than 0

$$\frac{N}{L'_2} \cdot T'_2 + \beta \cdot N \cdot \left(\frac{1}{L'_2} - \frac{1}{L_1}\right) \cdot T_2 < 0$$
$$\Leftrightarrow \frac{T'_2}{T_2} < \beta \cdot \left(\frac{L'_2}{L_1} - 1\right)$$

This result generalizes to inserting a new model at any position in a polybasic system. When inserting model $M_{new}$ between $M_i$ and $M_{i+1}$, we can treat all models before the insertion point ($M_1$ through $M_i$) as a single composite model, and all models after the insertion point ($M_{i+1}$ through $M_k$) as another composite model. This reduces the general case to the three-model case proven above, where the composite model before insertion corresponds to $M_1$, the new model corresponds to $M'_2$, and the composite model after insertion corresponds to $M_2$.

Therefore, the same conditions for efficiency improvement apply at any insertion point in the model sequence, subject to the specified constraints on acceptance lengths. $\square$

**Stability Analysis.** Beyond achieving higher acceptance lengths, stability in acceptance is crucial for consistent speedups. We analyze speculative sampling with probability $p_i = 1 - \alpha$ to accept a token from $M_{i+1}$ if it is likely under $M_i$'s distribution. Let $\sigma_i^2$ be the variance of acceptance lengths. As shown below, for multi-model chaining, acceptance length variance grows with smaller $p_i$, implying that high acceptance probability supports stable performance:

**Theorem 3.3** (Sampling Stability). *In the model chain using speculative sampling with acceptance probability $p_i = 1 - \alpha$, the variance in acceptance length satisfies:*

$$\sigma^2 = \frac{\alpha\left[1 - (n^2 - 1)\alpha^n\right] - (n^2 - 1)\alpha^{n+1}}{(1 - \alpha)^2}.$$

*Proof.* Let $p = 1 - \alpha$ be the probability of accepting a token. For a truncated geometric distribution of maximum $n$ trials, define:

$$S = \sum_{k=1}^{n-1} k \cdot (1 - p)^{k-1}.$$

Using standard manipulation (method of differences), one can show:

$$S = \frac{1 - (1-p)^{n-1} - n(1-p)^{n-1} + (1-p)^n}{p^2}.$$

Hence, the expectation of the acceptance length, allowing up to $n$ tokens, is

$$E[N] = \frac{1 - (1-p)^n}{p}.$$

We similarly compute

$$E[N^2] = \sum_{k=1}^{n-1} k^2 \cdot p \cdot (1-p)^{k-1} + n^2 \cdot (1-p)^{n-1}.$$

After careful algebra (omitted for brevity), one obtains:

$$E[N^2] =$$
$$\frac{1 - (1-p)^n(n^2 + 2n - 1) + 2(1-p)^{n+1}(n-1)}{p^2}.$$

Thus,

$$\text{Var}(N) = E[N^2] - (E[N])^2,$$

leading to the formula stated in Theorem 3.3. $\qquad\square$

Collectively, these results establish a principled foundation for polybasic speculative decoding. Given model inference times $T_i$ and acceptance probabilities, one can estimate the optimal system layout via Equation (3), gauge whether a new model confers net benefit (Theorem 3.2), and use speculative sampling to ensure stable acceptance lengths (Theorem 3.3).

### 3.3. Three-Model System Design

To illustrate practical deployment, we describe a three-model system ($M_1$: target, $M_2$: intermediate, $M_3$: lightweight) that exemplifies the design choices guided by our theory.

**Architecture.**   Our reference system includes:

- $M_1$ **(Target):** A high-capacity model such as Vicuna-7B or LLaMA2-Chat 7B.

- $M_2$ **(Intermediate):** A quantized 4-bit version of $M_1$ or a comparable mid-size model to bridge the capacity gap.

- $M_3$ **(Draft):** A lightweight, fast model (e.g., EAGLE2(Li et al., 2024)) for initial token proposals.

By Theorem 3.2, $M_2$ should be inserted if it raises the acceptance length enough to offset its own forward-pass cost.

**Staged Verification.**   Tokens first pass from $M_3$ to $M_2$, whose verification is relatively fast. Accepted tokens are then periodically verified by $M_1$. This two-stage verification acts as a filter, rapidly discarding problematic tokens at the cheaper $M_2$ stage. The threshold for passing tokens to $M_1$ is set to accumulate a small block (e.g., $\mu$ tokens) to amortize $M_1$'s forward-pass overhead. This setup capitalizes on the fact that $M_3$ can generate numerous tentative tokens quickly, while $M_1$ only checks consolidated blocks of already moderately validated tokens.

---

**Algorithm 1** Polybasic Speculative Decoding (Three Models)

---

1: **Input:** Target model $M_1$, intermediate model $M_2$, draft model $M_3$
2: **Input:** Context $x_{\leq t}$, total length $N$
3: **Input:** Draft length $K$, threshold $\mu$
4: **Initialize:** $t \leftarrow |x_{\leq t}|$, accepted $\leftarrow \emptyset$, cnt $\leftarrow 0$
5: **while** $t < N$ **do**
6:     // Draft and verify with $M_3$ and $M_2$
7:     $\widetilde{x}_{1:K} \leftarrow M_3(x_{\leq t})$    // Draft
8:     $p_{1:K} \leftarrow M_2(x_{\leq t}, \widetilde{x}_{1:K})$   // Verify
9:     **for** $i = 1$ **to** $K$ **do**
10:       **if** VERIFY($\widetilde{x}_i, p_i$) **then**
11:         accepted.append($\widetilde{x}_i$)
12:         cnt $\leftarrow$ cnt $+ 1$
13:       **else**
14:         **break**
15:       **end if**
16:     **end for**
17:     // Check if threshold reached for $M_1$ verification
18:     **if** cnt $\geq \mu$ **then**
19:       $v \leftarrow M_1(x_{\leq t}, \text{accepted})$ // Verify
20:       **if** VERIFYBLOCK(accepted, $v$) **then**
21:         $x_{t+1:t+\text{cnt}} \leftarrow$ accepted
22:         $t \leftarrow t + \text{cnt}$
23:       **else**
24:         $x_{t+1} \leftarrow$ SampleOne($v_1$) // fallback acceptance
25:         $t \leftarrow t + 1$
26:       **end if**
27:       accepted $\leftarrow \emptyset$
28:       cnt $\leftarrow 0$
29:     **end if**
30: **end while**

---

**Algorithm.**   Algorithm 1 details a generic procedure for polybasic speculative decoding with three models. The system accumulates tokens verified by $M_2$ until a threshold, then triggers verification by $M_1$. Upon acceptance or partial acceptance, it appends tokens to the growing output sequence and advances $t$. Simple or more sophisticated error handling (e.g., partial rollback) can be adopted if $M_1$ rejects tokens.

Such a staged design is representative rather than exhaustive. Model scaling, verification strategies, and drafting lengths can be adapted to different resource constraints or performance targets. Our theoretical framework offers explicit performance bounds, making the design space more transparent.

### 3.4. Generalization to Self-Drafting Methods

Our polybasic speculative decoding framework provides a general theoretical foundation that naturally extends to self-

drafting approaches, demonstrating the broad applicability of our theoretical principles. This generalization reveals new opportunities for designing efficient speculative decoding systems using model-internal components.

In self-drafting methods like FFN heads approaches (Cai et al., 2024) (Ankner et al., 2024) (Kim et al., 2024b) and early exiting (Elhoushi et al., 2024) techniques, multiple prediction sources are derived from the same model architecture. These can be viewed as the fundamental draft models in a polybasic system, with the original model serving as the target. For instance, in an FFN heads approach, each head functions as a base draft model capable of generating tokens in parallel, forming the lowest tier in our polybasic hierarchy. Similarly, early exit points at different layers can be treated as draft models with varying capabilities.

The optimization principles established in Section 3.2 remain applicable in this context, though they require careful algorithmic design to account for the shared computational paths in self-drafting approaches. The relationship between forward pass frequency and acceptance length provides guidance for optimal configuration of these systems, while our stability analysis (Theorem 3.3) informs the design of verification strategies.

This generalization demonstrates that our polybasic framework provides a unified theoretical foundation for speculative decoding, encompassing both independent draft models and self-drafting approaches. This suggests promising directions for developing new algorithmic techniques that fully exploit the parallel prediction capabilities inherent in these methods while maintaining the theoretical guarantees of our framework.

# 4. Experiments

We conduct comprehensive evaluations to validate our theoretical claims and to benchmark the proposed polybasic system against traditional dualistic strategies. Our experiments span various LLMs, tasks, and hyperparameter settings.

## 4.1. Setup and Metrics

**Models and Tasks.** We conducted experiments on Vicuna-7B, LLaMA2-chat-7B, and LLaMA3-7B-Instruct. We evaluated our multi-model speculative system in SpecBench(Xia et al., 2024), across multiple tasks including multi-turn conversation, translation, summarization, question answering, mathematical reasoning, and retrieval-augmented generation, employing the MT-bench (Zheng et al., 2023), WMT14 DE-EN, CNN/Daily Mail (Nallapati et al., 2016), Natural Questions (Kwiatkowski et al., 2019), GSM8K (Cobbe et al., 2021), and DPR (Karpukhin et al., 2020). Speculative sampling (Leviathan et al., 2023) conducted experiments with a batch size of 1, similarly, the majority of our experiments

also adopted this setting.

**Performance Metrics.** Following previous speculative decoding studies, we focus on two metrics. **Walltime speedup ratio** $c$: ratio of actual decoding time in our system vs. standard autoregressive decoding. **Average acceptance length** $\mu$: mean number of consecutively accepted tokens per forward pass by the final (largest) model.

**Quantization and Training Details.** For the intermediate model, we adopt 4-bit quantization (Ma et al., 2024) with a group size of 128, balancing reduced inference cost against quality. Draft models are built following EAGLE2, trained on ShareGPT data. Our experiments run on NVIDIA A800 80G GPUs.

## 4.2. Theoretical Validation

To empirically validate Theorem 3.2, we conducted two targeted experiments evaluating the impact of inserting additional draft models into a polybasic system. We measured $T_{\text{new}}$, $L_{\text{new}}$, and the resultant speedup ratio. Results are summarized in Table 4.2.

**Case 1: Non-Compliant Insertion** We inserted a lightweight Vicuna-1B model between Vicuna-7B (target) and EAGLE2 (baseline drafter). Here, $T_{\text{new}}/T_i = 0.80$, while the acceptance-length improvement factor $L_{\text{new}} \cdot (1/L_i - 1/L_{i\text{-new}}) = 0.117$. Since $0.80 > 0.117$, Theorem 3.2 predicts a performance degradation. Empirically, the speedup ratio dropped from $2.61\times$ to $1.08\times$, confirming the theoretical prediction. This highlights the necessity of balancing model capacity and computational overhead when expanding the polybasic hierarchy.

**Case 2: Compliant Insertion** We inserted a quantized Vicuna-7B (W4A16) model between the original Vicuna-7B and EAGLE2. Here, $T_{\text{new}}/T_i = 0.318$ and $L_{\text{new}} \cdot (1/L_i - 1/L_{i\text{-new}}) = 0.330$. Since $0.318 < 0.330$, Theorem 3.2 predicts a speedup improvement. Experimentally, the system achieved a $3.48\times$ speedup, up from $2.61\times$. This demonstrates the theorem's utility in guiding effective model selection.

**Case 3: Generalization** To substantiate the universal theoretical guidance of Theorem 3.2 across diverse cascaded speculative sampling methodologies, we reproduced Cascade Speculative Drafting (Chen et al., 2023b) and conducted rigorous evaluations spanning multiple model scales including FLAN-T5-XXL, Base, and Small variants. We inserted FLAN-T5-base between FLAN-T5-XXL and FLAN-T5-small. The configuration yields $T_{\text{new}}/T_i = 0.403$ with acceptance metric $L_{\text{new}} \cdot (1/L_i - 1/L_{i\text{-new}}) = 0.461$, satisfying the acceleration criterion $0.403 < 0.461$ as shown in Table 1. The system exhibits statistically significant speedup improvement from $3.19\times$ to $3.88\times$.

*Table 1.* Theoretical Validation via Model Insertion

| Case | $T_i$ (ms) | $L_{i\text{-new}}$ | $T_{\text{new}}$ (ms) | $L_{\text{new}}$ | $T_{i+1}$ (ms) | $L_i$ | **Speedup** |
|------|------|------|------|------|------|------|------|
| Non-compliant | 22 | 3.83 | 17.61 | 3.77 | 4 | 4.34 | $2.61\times \to 1.08\times$ |
| Compliant | 22 | 6.26 | 7.00 | 4.67 | 4 | 4.34 | $2.61\times \to 3.48\times$ |
| CS Drafting | 47.52 | 3.50 | 19.16 | 3.02 | 12.42 | 2.28 | $3.19\times \to 3.88\times$ |

These experiments directly corroborate Theorem 3.2, showing that the theoretical conditions on $T_{\text{new}}$ and $L_{\text{new}}$ are necessary for improving system performance. The results emphasize the framework's ability to rigorously guide model selection and hierarchy design, moving beyond heuristic-driven approaches.

### 4.3. Effectiveness

Figures 2 and 3 summarize speedup ratios on various tasks. Our polybasic approach demonstrates clear gains over dualistic baseline systems (including EAGLE2 and Speculative Sampling). Notably:

- **Vicuna-7B** achieves $3.16\times$ on average and up to $4.43\times$ in mathematical reasoning.

- **LLaMA2-Chat 7B** attains $3.66\times$ overall, peaking at $4.10\times$ in multi-turn conversation.

- **LLaMA3-8B** yields $3.31\times$–$3.87\times$ speedups, illustrating the method's adaptability to larger model sizes.

- **qwen2-7B-Instruct** demonstrates a $3.28\times$ average speedup, which is approximately 69% higher than EAGLE2's $1.94\times$ acceleration on the same model.

Our analysis also shows that average acceptance lengths range from 9.1 to over 10 tokens, significantly higher than typical dual-model methods. This corroborates our theoretical claim that multi-tiered speculation improves acceptance efficiency.

As shown in Figure 3, our method demonstrates substantial speedups across diverse tasks, with particularly strong performance in math reasoning (up to $4.43\times$) and multi-turn conversation (up to $4.10\times$). However, we observe relatively modest acceleration on summarization tasks, where the speedup ranges from $2.95\times$ to $3.41\times$. This pattern can be attributed to the higher token generation requirements in summarization, which necessitates maintaining KV caches across multiple models in our polybasic speculative decoding framework. Despite this limitation, our approach is orthogonal to KV cache optimization techniques, suggesting potential for further improvements through the integration of cache-focused methods.

### 4.4. Scalability to Larger Models

To demonstrate the generalizability of our framework across model scales, we conducted additional experiments with Vicuna-13B and LLaMA-2-chat-70B models. As shown in Table 3, our polybasic approach maintains significant advantages over EAGLE baseline even when scaled to larger models. Specifically, we achieve $2.69\times$ speedup with Vicuna-13B (vs. EAGLE2's $2.30\times$) and $2.92\times$ for LLaMA-70B (vs. $2.46\times$), while maintaining substantially higher average acceptance lengths. These results confirm that our method's benefits are not limited to smaller models but extend to larger-scale LLMs. The slightly reduced absolute speedup ratios compared to 7B models align with expectations, as larger models naturally incur higher verification costs that partially offset drafting efficiency gains.

### 4.5. Ablation Study: Speculative vs. Greedy Sampling

To validate the impact of speculative sampling on stability, we compare acceptance-length variance from speculative vs. greedy verification. We sample 50 queries in a three-model setup and record acceptance-length distributions (Figure 4). As anticipated, speculative sampling yields smaller variance, indicating more stable acceptance lengths across diverse inputs. This result aligns with Theorem 3.3 and further justifies the use of speculative sampling in multi-tier verification.

### 4.6. Four-Model System Discussion and Limitations

While our theoretical analysis suggests the potential benefits of incorporating more models into the polybasic speculative decoding framework, empirical implementation of systems with four or more models faces practical challenges. Under our sufficient (though not necessary) condition for model insertion efficiency, it is currently difficult to find suitable off-the-shelf models that satisfy the theoretical requirements without additional training. This limitation primarily stems from the stringent balance needed between acceptance length improvements and computational overhead. However, we believe this barrier is not fundamental. Through future exploration of complementary optimization techniques, such as advanced KV cache management, model pruning, and quantization, we anticipate achieving breakthroughs in systems with four or more models, potentially

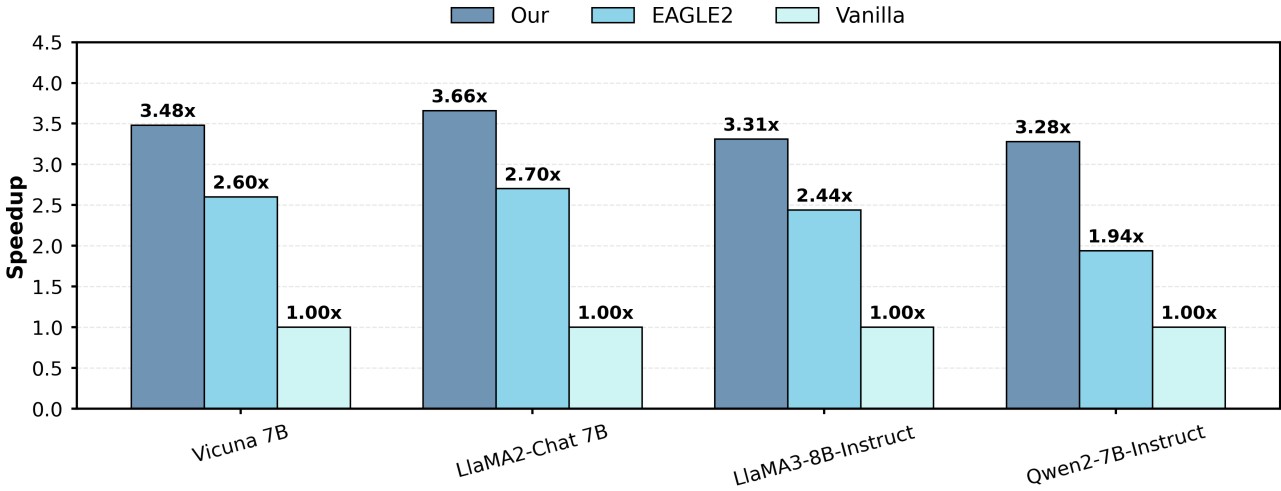

*Figure 2.* **Speedup ratios for Vicuna-7B, LLaMA2-Chat 7B, LLaMA3-8B-Instruct and Qwen2-7B-Instruct on SpecBench.** Our polybasic system consistently achieves the highest speedups ($3.16\times$–$3.66\times$), surpassing EAGLE2 and vanilla baselines.

*Table 2.* Average acceptance length ($\mu$) and speedup ratio ($c$) on different tasks. V7B: Vicuna-7B, L3-8B: LLaMA3-8B-Instruct, L2-7B: LLaMA2-Chat-7B, Q2-7B: Qwen2-7B-Instruct.

| | Model | MT $c$ | $\mu$ | Trans. $c$ | $\mu$ | Sum. $c$ | $\mu$ | QA $c$ | $\mu$ | Math $c$ | $\mu$ | RAG $c$ | $\mu$ | Overall $c$ | $\mu$ |
|---|---|---|---|---|---|---|---|---|---|---|---|---|---|---|---|
| | V7B | **3.77x** | **11.22** | **3.07x** | **7.76** | **3.01x** | **10.24** | **3.65x** | **9.53** | **4.43x** | **10.28** | **2.98x** | **10.30** | **3.48x** | **9.88** |
| Our | L3-8B | **3.70x** | **9.97** | **3.39x** | **8.86** | **3.02x** | **9.38** | **3.16x** | **9.08** | **3.87x** | **10.08** | **2.71x** | **9.24** | **3.31x** | **9.44** |
| | L2-7B | **4.10x** | **10.47** | **3.46x** | **9.15** | **3.41x** | **9.86** | **3.61x** | **9.49** | **4.02x** | **9.99** | **3.31x** | **10.08** | **3.66x** | **9.84** |
| | Q2-7B | **3.65x** | **9.85** | **3.15x** | **8.65** | **2.95x** | **9.15** | **3.25x** | **8.95** | **3.85x** | **9.95** | **2.85x** | **9.35** | **3.28x** | **9.32** |
| | V7B | 3.19x | 4.76 | 2.07x | 3.22 | 2.59x | 3.96 | 2.45x | 3.71 | 3.19x | 4.72 | 2.15x | 3.95 | 2.61x | 4.34 |
| EAGLE2 | L3-8B | 2.69x | 3.99 | 2.37x | 3.53 | 2.23x | 3.58 | 2.21x | 3.42 | 2.83x | 4.20 | 2.23x | 3.95 | 2.44x | 3.82 |
| | L2-7B | 3.04x | 4.48 | 2.61x | 3.96 | 2.50x | 4.04 | 2.55x | 4.05 | 3.04x | 4.68 | 2.40x | 4.19 | 2.70x | 4.30 |
| | Q2-7B | 2.40x | 3.74 | 1.45x | 2.45 | 1.59x | 3.06 | 1.81x | 2.91 | 2.63x | 4.26 | 1.72x | 3.27 | 1.94x | 3.51 |

*Table 3.* Speedup Ratios and Acceptance Lengths on Larger Models

| Method | Model | $c$ | $\mu$ |
|---|---|---|---|
| Our | Vicuna-13B | **2.69**$\times$ | **8.62** |
| | LLaMA-70B | **2.92**$\times$ | **7.48** |
| EAGLE | Vicuna-13B | 2.30$\times$ | 4.42 |
| | LLaMA-70B | 2.46$\times$ | 4.08 |

unlocking even greater acceleration benefits while maintaining inference quality.

Our polybasic framework, like other speculative decoding methods, can be limited by large KV cache footprints, which scale with text length. Thus, for tasks with extensive context, the overhead from additional models can be more pronounced. As Figure 3 and Table 2 shows, we observe somewhat lower acceleration in summarization and RAG tasks than in shorter contexts. Addressing KV cache constraints via caching techniques (Xiao et al., 2023; Zhang et al., 2024c;a; Jin et al., 2024) is an active research avenue and remains a promising direction for future improvements.

## 5. Conclusion

We have presented a *polybasic* speculative decoding system that systematically extends beyond dualistic draft-target paradigms. By establishing a rigorous theoretical framework, we derived an expression for optimal inference time and showed how speculative sampling stabilizes acceptance lengths in multi-model systems. Extensive experiments spanning multiple tasks and model families corroborate our claims, demonstrating $4\times$ speedups while preserving the

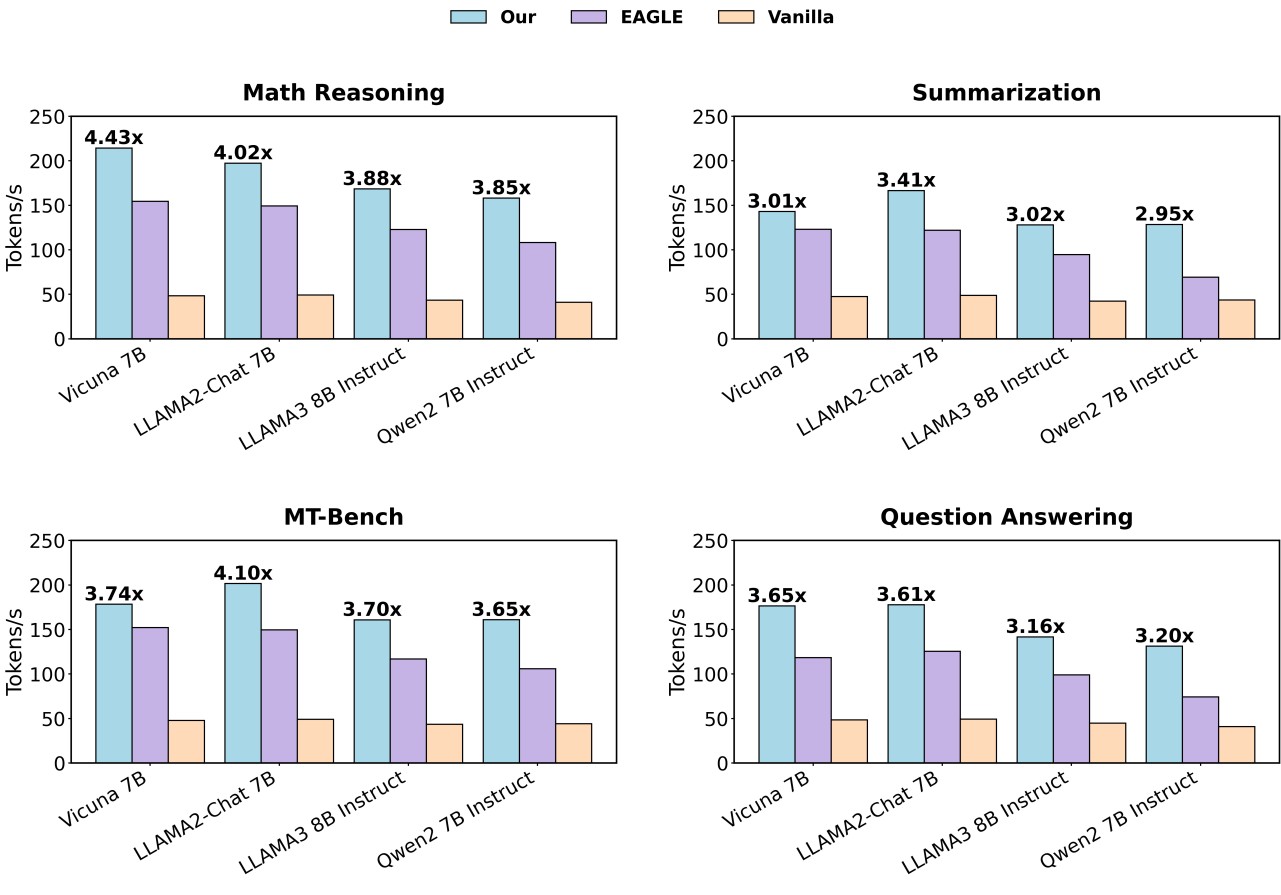

*Figure 3.* **Speedup by task.** Our method excels in math tasks, reaching $4.43\times$ with Vicuna-7B, while also maintaining strong accelerations in translation, QA, and multi-turn conversation.

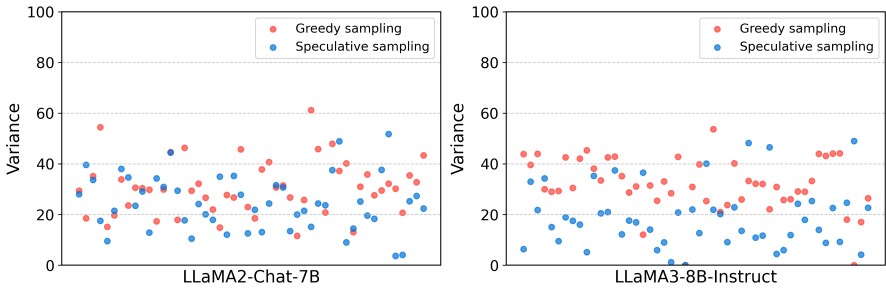

*Figure 4.* **Variance of acceptance length.** Speculative sampling (blue) exhibits noticeably lower variance compared to greedy sampling (orange), aligning with our theoretical stability analysis.

target model's output distribution.

In future work, we will extend our findings to more complex parallel computing scenarios by developing distributed speculative sampling systems. We also plan to explore more efficient caching strategies, implement dynamic adaptation of speculation lengths, and validate the framework's general-

ity across models of varying scales (from billions to trillions of parameters), aiming to continuously push the boundaries of efficient LLM inference.

## Acknowledgements

This work was supported by the National Science Fund for Distinguished Young Scholars (No.62025603), the National Natural Science Foundation of China (No. U21B2037, No. U22B2051, No. U23A20383, No. 62176222, No. 62176223, No. 62176226, No. 62072386, No. 62072387, No. 62072389, No. 62002305 and No. 62272401), and the Natural Science Foundation of Fujian Province of China (No. 2021J06003, No.2022J06001).

## Impact Statement

This paper presents work whose goal is to advance the field of Machine Learning. There are many potential societal consequences of our work, none which we feel must be specifically highlighted here.

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
