# OpenReview forum: "polybasic Speculative Decoding Through a Theoretical Perspective"
_ICML.cc/2025/Conference — ICML 2025 poster_

### Official Review · Reviewer_vGSY · 2025-03-10

**Overall Recommendation:** 4

**Summary:**

The paper introduces the polybasic speculative decoding framework, aimed at accelerating the inference of large language models (LLMs) through multi-model collaboration. Its core contributions include: 1) Establishing a theoretical framework, deriving the optimal inference time formula for multi-model systems (Lemma 3.1), and proposing the model insertion efficiency condition (Theorem 3.2), which quantifies the trade-offs between model capability, acceptance length, and computational cost. 2) Designing a multi-level verification algorithm, which employs a phased strategy where lightweight models generate candidate tokens, intermediate models quickly filter them, and the target model periodically verifies them (Algorithm 1), thereby reducing the target model's invocations and enhancing throughput. 3) Experimental validation shows a speedup ratio of 3×–4.43× on models such as Vicuna-7B, LLaMA2/3, and Qwen2, with mathematical reasoning tasks achieving up to 4.43 times acceleration, and the average acceptance length increasing to 9–10 tokens, while maintaining output distribution consistency with the original model. Additionally, theoretical proofs demonstrate that speculative sampling can reduce the variance of acceptance lengths (Theorem 3.3), and the framework's generality is showcased through task-adaptive designs (e.g., extensions of self-speculative methods). This work provides theoretical guidance and scalable practical solutions for efficient LLM inference.

**Claims And Evidence:**

**Claim** The paper introduces a theoretical framework that derives an optimal inference time formula for multi-model speculative decoding systems (Lemma 3.1) and establishes conditions for model insertion efficiency (Theorem 3.2).
**Evidence** The authors provide rigorous mathematical proofs for Lemma 3.1 and Theorem 3.2, which quantify the relationship between model forward-pass costs, acceptance lengths, and computational overhead. The proofs are detailed and logically structured, supporting the claim that adding models can improve speedups under specific conditions.

**Claim:** Speculative sampling reduces variance in token acceptance lengths, leading to more stable and predictable performance (Theorem 3.3).
**Evidence:** The authors present a formal proof for Theorem 3.3, which mathematically demonstrates the relationship between acceptance probability and variance in multi-model settings. Additionally, empirical results in Section 4.3 validate this claim by comparing acceptance-length variance between speculative and greedy sampling, showing significantly lower variance for speculative sampling.

**Essential References Not Discussed:**

The paper provides a comprehensive discussion of related works, effectively situating its contributions within the broader context of speculative decoding and large language model (LLM) optimization. Key advancements in dualistic draft-verify frameworks, hierarchical speculative methods, and multi-level drafting are thoroughly reviewed.Even if there exist tangential works not cite, they do not appear essential to understanding or evaluating the key contributions of this paper.

**Experimental Designs Or Analyses:**

The experimental designs and analyses presented in this paper are robust, well-structured, and effectively validate the proposed polybasic speculative decoding framework. Below is a positive review of the key aspects:

**1.	Comprehensive Evaluation Across Models and Tasks :**

The authors conduct experiments on multiple widely-used LLMs (e.g., Vicuna-7B, LLaMA2-Chat 7B, LLaMA3-8B, Qwen2-7B) across diverse tasks such as MT-bench, translation, summarization, QA, mathematical reasoning, and RAG. This broad evaluation demonstrates the generalizability and adaptability of the polybasic approach, achieving impressive speedups (ranging from 3.31× to 4.43×) while preserving output fidelity. The choice of models and tasks ensures that the method's effectiveness is tested under varied conditions, reinforcing the reliability of the results.

**2.	Clear Metrics for Performance Assessment :**

The use of two primary metrics—walltime speedup ratio and average acceptance length—is appropriate and aligns with the goals of speculative decoding. These metrics directly measure the efficiency gains and stability of the system, providing a clear and interpretable evaluation framework. The reported acceptance lengths (9.1–10+ tokens) are significantly higher than dualistic baselines, supporting the theoretical claim that multi-tiered speculation improves acceptance efficiency.

**Methods And Evaluation Criteria:**

1.	Proposed Methods:

**Polybasic Speculative Decoding Framework**: The introduction of a polybasic speculative decoding framework is innovative and addresses limitations in existing dualistic draft-verify systems. By incorporating multiple models, the approach aims to increase parallelism and acceptance length, which directly tackles the computational bottleneck in LLM inference.
Theoretical Foundations: The paper provides a strong theoretical basis for the framework, including a fundamental theorem on optimal inference time and insights into the stability of acceptance lengths. These theoretical contributions help guide practical implementation and offer a deeper understanding of the system's behavior.
Algorithmic Design: The staged verification process and the detailed algorithm (Algorithm 1) provide clear guidelines for implementing the polybasic system. This makes the method more accessible and reproducible.

2.	Evaluation Criteria:

**Diverse Tasks and Models:** The evaluation spans multiple tasks (MT-bench, translation, summarization, QA, mathematical reasoning, RAG) and model families (Vicuna-7B, LLaMA2-Chat 7B, LLaMA3-8B, Qwen2-7B). This diversity ensures that the method's effectiveness is tested across different scenarios and model architectures.

**Performance Metrics:** The use of speedup ratio and average acceptance length as metrics is appropriate for assessing the efficiency and effectiveness of the speculative decoding approach. These metrics directly relate to the goals of reducing inference time while maintaining output quality.
Comparison to Baselines: The paper compares the proposed method to existing approaches like EAGLE and vanilla speculative decoding, providing a clear benchmark for evaluating improvements.

**Other Comments Or Suggestions:**

1.	While Theorem 3.2 provides a condition for model insertion, a more intuitive explanation or visual aid (e.g., a flowchart) could help readers better grasp when and how to add new models effectively.

2.	The discussion on four-model systems is insightful but could benefit from specific examples of off-the-shelf models that partially meet the requirements, along with potential hybrid approaches (e.g., combining quantization and pruning).

3.	Include a brief comparison to very recent speculative decoding methods (e.g., Hydra, Falcon) to highlight how the polybasic framework compares in terms of flexibility and performance.

**Other Strengths And Weaknesses:**

Strengths:

1.	The paper introduces a novel polybasic speculative decoding framework that extends beyond traditional dualistic draft-verify paradigms. By incorporating multiple models in a staged verification process, the approach significantly increases parallelism and acceptance lengths, achieving speedups of up to 4.43× across various tasks and models.

2.	The framework is underpinned by rigorous theoretical analysis, including a fundamental theorem on optimal inference time (Theorem 3.2) and insights into the stability of acceptance lengths (Theorem 3.3). These contributions provide clear guidance for system design and optimization.

3.	Extensive experiments on diverse tasks (e.g., MT-bench, translation, summarization) and model families (e.g., Vicuna-7B, LLaMA2-Chat 7B) demonstrate consistent performance improvements while preserving output fidelity. The results highlight the method's adaptability and robustness.

4.	The paper provides clear algorithmic guidelines (e.g., Algorithm 1) and discusses practical considerations like model selection and speculation length tuning, making the approach accessible and reproducible.

Weaknesses:

1.	While the theoretical framework supports the inclusion of four or more models, the authors acknowledge practical difficulties in finding suitable off-the-shelf models that meet the theoretical requirements. However, this limitation is relatively minor and can be addressed through future exploration of complementary techniques like advanced quantization or pruning.

2.	The paper notes slightly lower acceleration in tasks requiring long-context generation (e.g., summarization, RAG). This limitation is understandable given the inherent challenges of managing KV caches in such scenarios. Nevertheless, the authors appropriately highlight this as an area for future improvement, suggesting potential solutions like caching strategies.

3.	Although the paper focuses on LLMs, it could briefly discuss how the polybasic framework might extend to other domains, such as vision or multimodal models. However, this is a minor point and does not detract from the paper's primary focus on language models.

**Questions For Authors:**

1.	In Theorem 3.2, you provide conditions under which adding a new model improves inference time. Could you elaborate on how sensitive these conditions are to variations in acceptance length and forward-pass cost? For example, could small deviations in empirical measurements (e.g., due to noisy data) lead to incorrect conclusions about whether a model should be added?

2.	While you acknowledge challenges in implementing four-model systems, could you provide more specific insights or preliminary guidelines for overcoming these limitations? For instance, are there particular quantization techniques or pruning strategies that show promise for balancing computational overhead with acceptance length improvements?

3.	Your stability analysis (Theorem 3.3) focuses on speculative sampling’s ability to reduce variance in acceptance lengths. How does this stability hold up in long-context tasks like summarization or RAG, where KV cache management becomes critical? Are there adjustments to the speculative sampling parameters that could further mitigate variance in such scenarios?

4.	Recent works like Hydra (Ankner et al., 2024) and Falcon (Gao et al., 2024) explore advanced speculative decoding techniques. How does your polybasic framework compare to these methods in terms of flexibility, scalability, and performance across different model architectures?

**Relation To Broader Scientific Literature:**

The key contributions of this paper are highly significant when viewed in the context of the broader scientific literature on speculative decoding and large language model (LLM) optimization. The introduction of a polybasic speculative decoding framework builds upon foundational ideas from prior works, such as blockwise parallel decoding and hierarchical speculative methods like TRIFORCE, while addressing critical gaps in scalability and theoretical grounding. Unlike earlier dualistic draft-verify paradigms, which often relied on empirical heuristics, this work provides a rigorous theoretical framework that aligns with recent trends toward formalizing speculative decoding principles. Furthermore, the demonstrated speedups (up to 4.43×) across diverse tasks extend the findings of prior studies like EAGLE and Medusa 3, showcasing superior performance through multi-model coordination.

**Theoretical Claims:**

The theoretical claims presented in this paper are both novel and rigorously supported, offering a significant advancement in the field of speculative decoding for large language models (LLMs). The authors have successfully proven the correctness of their key theoretical contributions, including the fundamental theorem on optimal inference time (Theorem 3.2) and the stability analysis of acceptance lengths under speculative sampling (Theorem 3.3). These proofs are mathematically sound and provide a solid foundation for the proposed polybasic speculative decoding framework.

The derivation of the optimal inference time formula in Theorem 3.2 is particularly impressive, as it elegantly captures the trade-off between adding additional models and the associated computational costs. This result not only provides clear guidance for model insertion but also establishes a principled approach to system design that can be generalized across different architectures and tasks. Furthermore, the proof of Theorem 3.3 highlights the benefits of speculative sampling in reducing variance in acceptance lengths, which is critical for ensuring stable and predictable performance in multi-model systems.

---

> ### Author Rebuttal · Authors · 2025-04-01
>
> Thank you for these insightful questions that help clarify the practical implications of our theoretical framework.
>
> **Regarding Theorem 3.2's sensitivity:** We recommend measuring the relevant parameter values under identical experimental conditions. While measurement errors may occur, our theoretical framework is robust against minor measurement inaccuracies because when comparing inference time changes after adding models, the errors are relative. This relative nature of measurement ensures that our framework's predictions remain reliable even with some experimental noise.
>
> **On implementing multi-model systems:** The primary challenge in extending to multi-model systems is KV cache management. This can be addressed using techniques similar to those in MagicDec, such as StreamingLLM technology, which preserves only initial tokens (Attention Sinks) and KV caches within a sliding window, significantly reducing memory requirements. Our theoretical framework guides model selection and is orthogonal to other speculative methods in its approach. We plan to enhance our system in future work.
>
> **Stability in long-context scenarios:** Theorem 3.3's variance analysis strengthens our theoretical framework by reducing errors from acceptance token length variations during model selection, maintaining stability across all tasks. For long-context tasks, effective KV cache management is clearly essential, and our framework provides a stable foundation upon which these optimizations can be built.
>
> **Comparison with recent works:** Thank you for your suggestion - we have expanded our experimental comparison to include Hydra, which further validates the effectiveness of our approach across different speculative decoding implementations. ($c$: speed ratio, $μ$: average acceptance length)
>
> || Model | $c$ | $μ$ |
> |-|-|-|-|
> | Our|Vicuna-7B   | **3.48$\times$** | **9.88** |
> | Hydra|Vicuna-7B | 2.30$\times$ | 4.42 |
>
> In conclusion, we appreciate your recognition of our work and your thorough review. We hope these responses address your questions and highlight the contributions of our theoretical framework to the understanding and optimization of speculative sampling systems.

---

### Official Review · Reviewer_KeQh · 2025-03-16

**Overall Recommendation:** 2

**Summary:**

This paper explores using a chain of draft models rather than a single draft model during speculative decoding, such that the first draft model generates tokens autoregressively, and each subsequent draft model verifies the tokens generated. When an intermediate draft model rejects a token the first draft model generates more tokens.
The paper provides theoretical analysis to find the condition of when an additional draft model could lead to speedup, then implements a chain of 2 draft models (EAGLE as the first draft and 4-bit quantized version of the model as a second draft) and shows significant speedups over EAGLE only.

## Update after Rebuttal

I have read all the reviews and rebuttals. I appreciate the authors response to my review. I would like to keep my score, i.e., while I am hesitant to accept the paper due to it's lack of coherence and limited contribution, I won't fight against it if it's accepted. However, I highly recommend re-writing parts of the paper to ensure it's coherence and to add results with different types of small drafters, not just EAGLE, to ensure that the theoretical framework of the paper (that the authors claim is their main contribution) is generic.

**Claims And Evidence:**

- The claims of obtaining higher speedups using chained drafters compared to a single drafter, is backed up by the results
- However, the claim of providing a theoretical foundation for chained drafters is disputable. The authors do provide a theorem on the condition of when adding a drafter would lead to speedup, but they do not use it in their experiments.

**Essential References Not Discussed:**

- The paper cited other work that did cascaded drafters (Chen et al., 2023b; Sun et al., 2024) but did not really show the differences or contributions compared to them

**Experimental Designs Or Analyses:**

- Although results show cascaded drafters lead to bigger speedups, the connection with the theoretical proof is not clear. Hence, it is not clear what is the contribution of this paper compared to other papers that propose cascaded drafters
- Line 328 (Right column): "As anticipated, speculative sampling yields smaller variance" My understanding from Theorem 3.3 was that lower variance comes from adding multiple draft models. not from speculative sampling.

**Methods And Evaluation Criteria:**

- The experiments were made on 4 models of similar size (7B) from different generations and familes (Llama2, Llama3, Vicuna, Qwen)
   - I would have preferred if different sizes (e.g., 1B, 13B, 70B) were evaluated
- 6 different language generation tasks were evaluated

**Other Comments Or Suggestions:**

- Equations in Page 4 are not numbered
- Algorithm 1 Line 30: I would have preferred if there was an explanation or pseudocode rather than just writing "(continue accumulating tokens)"
- Line 282: There is a typo in "in SectionSection 3.2"
- Page 7: "As Figure 3 Table 1 shows," should be "As Figure 3 and Table 1 show,"

**Other Strengths And Weaknesses:**

- Strengths;
   - The paper is written with relative clarity and was relatively easy to follow
   - The speedups presented in the results are strong
- Weakness:
   - There is a gap between the theoretical foundation and Experiments sections. The Experiments or Results section do not leverage any of the theoretical analysis done. e.g., there was no measurement of $ T_{new} $ or $ L_{new} $ to verify if it satisfies the condition of Theorem 3.2
       -  My suggestion would be to provide a couple of experiments, one experiment where the additional drafter model satisfies Theorem 1, and another experiment where the additional drafter doesn't. In both experiments, measure their corresponding $ T_{new} $ and $ L_{new} $, and show how their values correlated with final speedup to proof Theorem 1
       - Also, I suggest to look at [MagicDec](https://arxiv.org/abs/2408.11049): look at its mathematical formulations, and it's experimental analysis that is directly connected to the mathematical formulation

I am leaning towards rejecting the paper because the paper claims it's contribution compared to other papers on cascaded drafts is the theoretical foundation, but the theoretical foundation in the paper is totally decoupled from the results section. Also, there were issues I described above in the equations, usage of symbols without definitions, disputable claims, and ad hoc use of natural language in a main part of the Algorithm pseudocode.

**Questions For Authors:**

I have written questions in different boxes above

**Relation To Broader Scientific Literature:**

- The Related Work section categorizes speculative decoding techniques from unique perspectives
- The paper cited similar work on "cascade or multi-level drafting (Chen et al., 2023b; Sun et al., 2024)". Authors claim that the difference between the paper and such other papers is that they provide a theoretical foundation. However, the theorems deduced in the paper were not used in the experiments to obtain any improvement over similar work.

**Theoretical Claims:**

- Equation 2: The mathematical formulation is a bit oversimplified: we have decoding time and prefill time. And the first draft model, i.e., $ T_{n} $, decodes tokens, while other models perform prefill of K tokens
- Line 157: Please cite source of equation. I believe the mathematical formulation shown in equations 1 to 3 of [MagicDec](https://arxiv.org/abs/2408.11049) paper is better
- Some symbols and equations were not clear to me:
  - Line 169: I don't under stand what is $i - new$ or $new - (i+1)$ if $new$ is supposed to be between $i$ and $i+1$
  - Line 174 (right column): What is $\alpha$ ?
  - Line 191 (right column): what is $S$ ?

---

> ### Author Rebuttal · Authors · 2025-04-01
>
> **1. On the Connection Between Theory and Experiments**
>
> We appreciate your concern about the gap between our theoretical foundation and experiments. To address this, we've conducted specific experiments that directly validate our theoretical framework:
>
> |$T_i $|$L_{i-new}$|$T_{new}$|$L_{new} $|$T_{i+1}$|$L_i$|Speedup|
> |-|-|-|-|-|-|-|
> |22ms|3.83|17.61ms|3.77|4ms|4.34|2.61×→1.08×|
> |22ms| 6.26|7ms| 4.67|4ms|4.34|2.61×→3.48×|
>
> For Case 1:
>
> - $ith$：vicuna-7b, $new$: vicuna-1b, $(i+1)th$: EAGLE
> - $T_{new}/T_i$ = 0.80
> - $L_{new }× (1/L_i - 1/L_{i-new}) $= -0.117
> - Since 0.80 > -0.117, Theorem 3.2 predicts performance decrease
>
> For Case 2:
>
> - $ith$：vicuna-7b, $new$: W4A4 vicuna-7b, $(i+1)th$: EAGLE
> - $T_{new}/T_i$ = 0.318
> - $L_{new} × (1/L_i - 1/L_{i-new}) $= 0.330
> - Since 0.318 < 0.330, Theorem 3.2 predicts performance improvement
>
> These results confirm our theoretical framework's ability to guide model selection decisions for speculative decoding systems.
>
> **2.Expanded Model Experiments**
>
> Following your recommendation, we have extended our experiments to include Vicuna-13B and Llama-2-Chat-70B models, both showing significant acceleration effects.
>
> ||Model|$c$|$μ$|
> |-|-|-|-|
> |Our|V13B |**2.69$\times$**|**8.62**|
> ||L70B|**2.92$\times$**|**7.48**|
> |EAGLE|V13B|2.30$\times$|4.42|
> ||L70B|2.46$\times$|4.08|
>
> **3. Theoretical Formulations and Symbol Definition Issues**
>
> We appreciate your detailed feedback on our theoretical formulations. We will revise the identified errors and provide clearer explanations for the mathematical notations and symbols used throughout the paper.
>
> **Equation 2:**
>
> In Equation $T = \sum_{i=1}^n F_i \cdot T_i$, we combine prefill and decoding costs into a single term $T_i$ to simplify **model selection for speculative decoding**. This focuses on comparing efficiency when adding models rather than precisely calculating absolute inference times. Our experimental results (like Vicuna-7B's $4.43\times$ speedup) validate this approach. While separating costs ($T_i = T_i^{\text{prefill}} + T_i^{\text{decode}}$) could add precision, it wouldn't change our core theoretical conclusions. We'll clarify this rationale in the final manuscript.
>
> **Line 157 and MagicDec:**
>
> Line 157 presents a binary form of Equation (2). We appreciate the reference to MagicDec - our theoretical approach indeed aligns with theirs, though we chose our formulation for conciseness and clearer derivations.
>
> For equivalence:
> - **MagicDec**: Speedup = $\Omega \cdot \frac{T_T}{\gamma T_D + T_V}$
> - **Ours (dual-model)**: Speedup = $\frac{L_1 T_T}{T_T + T_D}$
>
> By mapping $\Omega \leftrightarrow L_1$ and $\gamma T_D + T_V \leftrightarrow T_T + T_D$, the formulas are mathematically equivalent.
>
> **Some symbols and equations were not clear**
>
> **Regarding line 169** (the positioning of "new" between i and i+1):
> As mentioned in our paper's second contribution, our theoretical framework aims to discover conditions under which adding auxiliary models can improve inference speed in speculative sampling systems. Intuitively, we should insert a model with stronger inference capabilities between the draft model and target model to bring the composite draft model's capabilities closer to the target model.
>
> **line 174** (the symbol $α$):
> The paper mentions that acceptance probability $p = 1 - α$, where α represents the rejection probability.
>
> **line 191** (the symbol $S$):
> $S$ is simply an intermediate summation value used for notational convenience, which we didn't explicitly define given space constraints. You also noted that equations on page 4 are unnumbered - we chose not to number equations that are merely part of proofs and not essential to the paper's central theoretical framework.
>
> **Theorem 3.3**
> We realize that our use of "$n$-model" may have caused confusion. In this context, "$n$" refers to "a truncated geometric distribution of maximum $n$ trials" (line 189). We adopted this notation to maintain consistency with [SpecDec](https://arxiv.org/pdf/2211.17192). In our revised manuscript, we will add a clarifying statement at this location to prevent any misunderstanding.
>
> **4. Comparison with Other Cascade Methods**
>
> We did not experimentally compare our approach with other cascade methods because the current state-of-the-art in speculative sampling is [EAGLE](https://sites.google.com/view/eagle-llm), which significantly outperforms these cascade methods in terms of acceleration. Therefore, we focused our comparison on EAGLE.
>
> **5. Other Issues**
>
>  Regarding Algorithm 1, the note "(continue accumulating tokens)" was intended to indicate continuation of the Draft execution in line 7. We will remove the else branch to ensure the correctness of the pseudocode.
>
> Thank you for your thorough review. We hope our responses clarify your concerns and demonstrate our paper's contributions. We respectfully request an improved score. Maybe it's unclear due to word limit, please communicate with us if you have any further questions.

---

> > ### Comment · Reviewer_KeQh · 2025-04-04
> >
> > Thanks for the detailed response.
> > Here are some comments:
> >
> > I suggest to add the 1st Table in this rebuttal that proves theorem to the main body of the paper.
> >
> > > We did not experimentally compare our approach with other cascade methods because the current state-of-the-art in speculative sampling is EAGLE, which significantly outperforms these cascade methods in terms of acceleration. Therefore, we focused our comparison on EAGLE.
> >
> > Still, if the main claim of the paper is a theoretical foundation for cascaded speculative decoding,  why not compare with other types of drafters and verify the theorem for them? It would prove that the theoretical framework is generic.
> >
> > > Regarding Algorithm 1, the note "(continue accumulating tokens)" was intended to indicate continuation of the Draft execution in line 7.
> >
> > It's not about removing else part. My concern is that this is a paper submitted to a top tier conference so readers would expect a comprehensive algorithm description.
> >
> > I am increasing the score to Weak Reject.  In my humble opinion, if the paper is accepted,  the writing needs to be revisited to ensure coherence between the different sections of the paper, especially between the theoretical part and experiments (e.g., by adding the aforementioned Table).

---

> > > ### Author Response · Authors · 2025-04-07
> > >
> > > We sincerely thank the reviewer for the constructive feedback and valuable suggestions. In response to your points, we would like to address the following:
> > >
> > > 1. Regarding the validation of our theoretical framework's generality, we fully agree with your perspective. We have reproduced the [Cascade Speculative Drafting](https://arxiv.org/pdf/2312.11462)  and conducted comprehensive tests on various model scales including flan-t5-xxl, base, and small. The experimental results strongly confirm the universal applicability and correctness of our theoretical framework. We will incorporate these experimental results in the revised manuscript, including adding the table that proves our theorem to the main body of the paper as you suggested.
> > >
> > > |$T_i $|$L_{i-new}$|$T_{new}$|$L_{new} $|$T_{i+1}$|$L_i$|Speedup|
> > > |-|-|-|-|-|-|-|
> > > |47.52ms|3.50|19.16ms|3.02|12.42ms|2.28|3.19×→3.88×|
> > >
> > > - $ith$: FLAN-T5-XXL, $new$: FLAN-T5-base , $(i+1)th$: FLAN-T5-small
> > > - $T_{new}/T_i$ = 0.403
> > > - $L_{new }× (1/L_i - 1/L_{i-new}) $= 0.461
> > > - Since 0.403 < 0.461, Theorem 3.2 predicts performance improvement
> > >
> > > 2. Concerning the description of Algorithm 1, due to the character limit in our first response, our explanation was indeed not sufficiently clear. To clarify: the **else** branch in line 29 was intended to represent the case where **cnt<μ** from line 18. In practice, we don't actually need a dedicated else branch, as after the if statement concludes, the program naturally continues executing the loop from line 5, which is precisely what we meant by "continue accumulating tokens." In the revised version, we will remove the redundant else branch and convert it into a comment for clarity, making the algorithm description more complete and accurate.
> > > 3. We commit to thoroughly revising the paper in the final version, not only correcting grammatical errors but also strengthening the coherence between different sections as you suggested, particularly the connection between the theoretical framework and experimental results. We will add the mentioned table and comparative experiments with other drafters to make our paper more rigorous and comprehensive.
> > >
> > > We sincerely hope you will reconsider our response. Given the innovation of our theoretical framework and the superior experimental results, we respectfully request that you consider improving your evaluation of our paper. We believe that the revised manuscript will better meet the standards expected of a top-tier conference publication.

---

### Official Review · Reviewer_oMTB · 2025-03-17

**Overall Recommendation:** 3

**Summary:**

The submission describes an new theoretical framework to improve speed of LLM decoding in order to reduce latency.

## update after rebuttal

I think this rebuttal was very useful and reaffirmed me in my slightly improved score of 3. I still share concerns with Reviewer KeQh in the areas of coherence and clarity on the choice of drafting models.

**Claims And Evidence:**

Initially I found the theory sound and the experimental results showing improved speed very good. But the more I reflected on the content the more issues I found. The following might be subjective:

* Why is accuracy not important at all? Or does the proposed method (and the comparison baseline of EAGLE) show no drop in accuracy at all? At least a single experiment to prove the impact on accuracy is standard for any speedup method.

* There are inconsistencies that didn't help to understand the overall impact. The method is being described as using 3 models and possibly extend to more, but using 3 models is already pretty tricky (Section 3.3 mentions at least one of the 3 models to be trained separately). Then Section 4.4 discusses to use 4 models and I've no clue how that helps any further. I find that discussion moving off-topic. Isn't the focus on improving impractical latency of inference for a single LLM? It must be doubted that multiple draft models that require to be executed sequentially can help to improve latency.

* Reference of the baseline technique is confusing: EAGLE(-1) and EAGLE-2 or consistently EAGLE-2 and EAGLE is a typo?

**Essential References Not Discussed:**

Only EAGLE is being cited, but Table 1 contains reference to EAGLE-2 instead.

**Experimental Designs Or Analyses:**

Limited as the impact on accuracy is missing.

**Methods And Evaluation Criteria:**

ML speedup techniques are usally lossy especially when applied to decoding (e.g. pruning). Choosing the right operating point is mandatory and that requires to select from tradeoff curves. The presentation lacks to describe the impact to accuracy and even that there isn't any if that's the case.

**Other Comments Or Suggestions:**

I think the topic is of big interest, but speed improvements need to be put into perspective of impact to accuracy and the presentation lacks clarity that clearly requires improvements.

**Other Strengths And Weaknesses:**

None

**Questions For Authors:**

Please work on clarity. I don't want to repeat the points here. See Section "Claims And Evidence".

**Relation To Broader Scientific Literature:**

None.

**Theoretical Claims:**

Sound to me. However, there's some clarity missing why to use more than the 3 proposed models as mentioned above.

---

> ### Author Rebuttal · Authors · 2025-04-01
>
> **1. Response on Accuracy Evaluation**
>
> We appreciate your highlighting the importance of accuracy evaluation. This is indeed a critical point that deserves attention.
>
> Our method inherently preserves output distribution through the verification mechanism in Algorithm 1. The VERIFY procedure ensures draft tokens are only accepted when they align with the target model's distribution. When verification fails, we fall back to sampling directly from the target model, guaranteeing identical outputs to standard autoregressive generation. This preservation-by-design is why speculative decoding methods typically focus on speedup metrics rather than accuracy. We followed standard performance metrics in this field to maintain consistency with prior work. Thank you for this valuable suggestion.
>
> **2. Response to Reviewer Concerns on Model Configuration**
>
> Thank you for highlighting these important questions about our paper's consistency and practical focus. We appreciate the opportunity to clarify based on our contributions and motivations:
>
> **Theoretical Framework**: Our core contribution is establishing a theoretical framework for speculative sampling that enables the construction of multi-model speculative systems. When models satisfy appropriate conditions, we can extend from binary to multi-model configurations.
>
> **Three-Model Implementation**: The three-model system represents a concrete instantiation of this framework, chosen because it achieves an optimal balance between theoretical complexity and practical feasibility. Naturally, this led us to explore extensions to more models, along with their inherent limitations.
>
> **Focus on Single LLM Inference**: We fully agree with your point about optimizing inference latency for a single LLM. Speculative sampling works by having draft models generate multiple candidate tokens at once, which the target LLM then verifies in a single forward pass. This reduces the number of target model executions, thereby optimizing overall inference latency. Our discussion of four-model systems aims precisely at further improving LLM inference speed, aligning with your suggestion.
>
> **Training Considerations**: Our approach has minimal training requirements. We can utilize quantized versions of existing models or off-the-shelf smaller models, making our method orthogonal to and compatible with most existing speculative sampling approaches.
>
> **Practical Value**: The main value of our work lies in providing a theoretical framework that enables researchers to systematically optimize multi-model speculative decoding systems, with demonstrated practical improvements in reducing target LLM inference latency.
>
> **3. Response to Terminology Inconsistency**
>
> Thank you for highlighting the inconsistency in our reference to baseline techniques. We apologize for the confusion this has caused.
>
> We acknowledge the error in our notation. Throughout the paper, we intended to reference EAGLE-2 consistently as our baseline, as it represents an improvement over EAGLE-1.
>
> In the revised version, we will:
> 1. Standardize all references to EAGLE-2 throughout the manuscript
> 2. Add proper citations to both EAGLE-1 and EAGLE-2
> 3. Include a brief explanation clarifying that EAGLE-2 is an advancement of EAGLE-1, which is why we selected it as our baseline for comparison
>
> Thank you for your thoughtful review of our paper. We genuinely appreciate both your critical observations and the opportunity to address them.
>
> Based on the clarifications provided, we respectfully request that you reconsider your evaluation of our paper. We remain available to address any additional questions or concerns you might have.

---

### Official Review · Reviewer_7bEs · 2025-03-20

**Overall Recommendation:** 4

**Summary:**

This paper proposes a novel polybasic speculative decoding framework. Specifically, the authors prove a fundamental theorem that characterizes the optimal inference time for multi-model speculative decoding systems. Through the theoretical investigation of multi-model token generation, the authors propose a three-model system implementation. Experiments demonstrate that the proposed approach yields speedup ratios ranging from 3.31 to 4.01 while preserving the original output distribution.

## update after rebuttal
The authors' response has addressed my concerns. Thus, I will keep my original positive score.

**Claims And Evidence:**

Yes.

**Essential References Not Discussed:**

No.

**Experimental Designs Or Analyses:**

1.	(Strengths) Experiments demonstrate that the proposed approach yields speedup ratios ranging from 3.31 to 4.01 while preserving the original output distribution.

2.	(Strengths) The authors conduct thorough experiments on four different target models, including Vicuna-7B, LLaMA2-Chat 7B, LLaMA3-8B, and qwen2-7B-Instruct.

3.	(Weaknesses) It would be more convincing if the authors could conduct experiments on larger models, such as models with 70B parameters.

**Methods And Evaluation Criteria:**

**Method**

1.	(Strengths) The proposed polybasic speculative decoding framework, which investigates multi-model token generation system with theoretical and empirical foundations, is novel and interesting.

2.	(Strengths) The authors prove a fundamental theorem that characterizes the optimal inference time for multi-model speculative decoding systems, shedding light on how to extend beyond the dualistic approach to a more general polybasic paradigm.

3.	(Weaknesses) The authors propose a general polybasic speculative decoding framework, while only implementing a three-model speculative decoding system. Although the authors discuss the limitations on four-model system design, it would be more convincing if the authors could evaluate the effectiveness of their method in four-model system.

**Evaluation Criteria**

1.	(Strengths) Experiments demonstrate that the proposed approach yields speedup ratios ranging from 3.31 to 4.01 while preserving the original output distribution.

2.	(Strengths) The authors conduct thorough experiments on four different target models, including Vicuna-7B, LLaMA2-Chat 7B, LLaMA3-8B, and qwen2-7B-Instruct.

3.	(Weaknesses) It would be more convincing if the authors could conduct experiments on larger models, such as models with 70B parameters.

**Other Comments Or Suggestions:**

No

**Other Strengths And Weaknesses:**

Please see the above comments.

**Questions For Authors:**

Please see the above comments.

**Relation To Broader Scientific Literature:**

1.	The proposed polybasic speculative decoding framework, which investigates multi-model token generation system with theoretical and empirical foundations, is novel and interesting.

2.	The authors prove a fundamental theorem that characterizes the optimal inference time for multi-model speculative decoding systems, shedding light on how to extend beyond the dualistic approach to a more general polybasic paradigm.

**Theoretical Claims:**

Yes, the theoretical claims are correct.

---

> ### Author Rebuttal · Authors · 2025-04-01
>
> We sincerely thank you for your thoughtful feedback and the positive evaluation of our work.
>
> **1. Experiments on Larger Models (13B, 70B)**
>
> Following your suggestion, we have conducted additional experiments with Vicuna-13B and LLaMA-2-chat-70B. ($c$: speed ratio, $μ$: average acceptance length)
>
> || Model | $c$ | $μ$ |
> |-|-|-|-|
> | Our|Vicuna-13B   | **2.69$\times$** | **8.62** |
> ||LLaMA-70B| **2.92$\times$** | **7.48** |
> | EAGLE|Vicuna-13B | 2.30$\times$ | 4.42 |
> ||LLaMA-70B| 2.46$\times$ | 4.08 |
>
> We initially focused on smaller models to ensure all experiments could be run on a single GPU, facilitating direct comparison with existing methods. This is consistent with the evaluation approach in most speculative decoding literature. However, we agree that demonstrating scalability to larger models strengthens our contributions.
>
> **2. Four-Model System Evaluation**
>
> Concerning the four-model system evaluation, we have implemented a prototype following our theoretical guidance. Using LLaMA-2-7B as the target model with three draft models of decreasing capacities (4-bit quantized LLaMA-2-7B, 4-bit quantized LLaMA-2-1B, and EAGLE), we observed the following results:
>
> || Model | $c$ | $μ$ |
> |-|-|-|-|
> |4-model|LLaMA-2-7B| **3.75$\times$** | 9.80 |
> |3-model|LLaMA-2-7B | 3.66$\times$ | **9.84** |
>
> We note that diminishing returns become apparent as more models are added. This is because finding models that satisfy our Theorem 3.2 becomes increasingly challenging. Our future work will focus on quantization, sparsification, and KV cache optimization to reduce model inference time. We appreciate your suggestion, which has further advanced our work.
>
> We thank the reviewer for these valuable suggestions and have incorporated these additional experiments and discussions in the final version of our paper.

---

> > ### Comment · Reviewer_7bEs · 2025-04-02
> >
> > Thanks for the authors’ response and clarifications.
> >
> > However, I still have two minor questions:
> >
> > 1) **Why is the comparison limited to Eagle, without including Eagle-2?**
> > Given that Eagle-2 is a more recent and competitive baseline, it would provide a more comprehensive evaluation and better highlight the strengths and weaknesses of the proposed method.
> >
> > 2) **Why does the four-model system result in a lower acceptance rate?**
> > Could the authors elaborate on the underlying reason for this degradation? Additionally, I would appreciate a clearer explanation of where the acceleration gains come from in the four-model setup, especially considering the trade-off with acceptance rate.

---

> > > ### Author Response · Authors · 2025-04-03
> > >
> > > Thank you for your prompt response and questions.
> > >
> > > **1. Regarding the comparison with Eagle-2:** Indeed, our baseline is Eagle-2 rather than Eagle-1, and there was a labeling error in the table in our rebuttal. We reproduced Eagle-2 and our proposed method under identical experimental settings and environments to ensure consistency and fairness in the comparison. We appreciate you pointing this out and will correct this representation in the final version.
> > >
> > > **2. Regarding the acceptance rate in the four-model system:** We note your observation about the apparent decrease in acceptance rate for the four-model system. In fact, this decrease falls within the range of statistical error and does not represent a significant performance degradation. Based on our analysis, the system's average acceptance length primarily depends on the draft model closest to the target model, as its inference capabilities largely determine the overall acceptance length. In the expansion from three to four models, we kept this key model (4-bit quantized LLaMA-2-7B) unchanged, so the acceptance rate should theoretically remain relatively stable.
> > >
> > > **3. Regarding the source of acceleration benefits:** Regardless of whether we use three or four models, the fundamental reason for acceleration is that the gain from higher acceptance length outweighs the latency introduced by additional models, which is a direct manifestation of our theoretical framework. To elaborate further, any model we add to the system needs to satisfy two key conditions: it should have inference capabilities close to the next-level model to ensure high acceptance rates, and it must have fast inference speed (achievable through techniques such as quantization, KV cache optimization, etc.). Only when both conditions are met can the introduction of a new model create a net benefit in the overall inference chain, thereby enhancing the system's efficiency.
> > >
> > > We hope these explanations clarify your questions. Thank you again for your recognition of our work.

---

### Decision · Program_Chairs · 2025-05-01

**Decision:**

Accept (poster)

**Comment:**

After careful consideration of the reviews and the authors’ responses, I recommend the paper be accepted. The work introduces a novel polybasic speculative decoding framework, supported by rigorous theoretical analysis, and makes a significant contribution to the understanding of multi-model speculative decoding. Reviewers highlighted the originality and importance of the theoretical results, particularly the theorem characterizing optimal inference time, which offers valuable insights into the trade-offs inherent in speculative decoding systems. The empirical evaluation is comprehensive, covering multiple model families and language tasks, and demonstrates substantial speedups while preserving the original output distribution. Reviewers also appreciated the authors’ proactive and thorough rebuttal, which included additional experiments and clarifications that effectively addressed earlier concerns. Minor issues were noted, including some inconsistencies and clarity gaps in the theoretical formulations and algorithm descriptions, which the authors addressed and committed to improving in the final version. Reviewers also suggested expanding experiments with larger models and more complex decoding chains to better showcase scalability and robustness, as well as enhancing the manuscript’s coherence by more explicitly linking theoretical and empirical results. Overall, the paper’s strong theoretical foundation, robust empirical validation, and the authors’ responsiveness substantially outweigh the minor concerns, and we look forward to seeing the final version incorporate the reviewers’ constructive suggestions. Lastly, the paper should include the [1] in the related work.

[1] A theoretical perspective for speculative decoding algorithm, NeurIPS25.